biomaterials/biomedical engineering/ microbiology

*Pseudomonas aeruginosa*, cyclic-di-GMP, biofilm, PilY1, poly(ethylene glycol) diacrylate, confocal microscopy

**Author for correspondence:**
Vernita D. Gordon
e-mail: gordon@chaos.utexas.edu

# Quantitative confocal microscopy and calibration for measuring differences in cyclic-di-GMP signalling by bacteria on biomedical hydrogels

Jacob Blacutt[1,2], Ziyang Lan[3], Elizabeth M. Cosgriff-Hernandez[3] and Vernita D. Gordon[1,2,4,5]

[1]Institute for Cellular and Molecular Biology, [2]Center for Nonlinear Dynamics, [3]Department of Biomedical Engineering, [4]Department of Physics, and [5]LaMontagne Center for Infectious Disease, The University of Texas at Austin, Austin, TX, USA

JB, 0000-0001-7905-253X; ZL, 0000-0001-7935-137X; EMC-H, 0000-0002-4701-5600; VDG, 0000-0002-5989-9378

The growth of bacterial biofilms on implanted medical devices causes harmful infections and device failure. Biofilm development initiates when bacteria attach to and sense a surface. For the common nosocomial pathogen *Pseudomonas aeruginosa* and many others, the transition to the biofilm phenotype is controlled by the intracellular signal and second messenger cyclic-di-GMP (c-di-GMP). It is not known how biomedical materials might be adjusted to impede c-di-GMP signalling, and there are few extant methods for conducting such studies. Here, we develop such a method. We allowed *P. aeruginosa* to attach to the surfaces of poly(ethylene glycol) diacrylate (PEGDA) hydrogels. These bacteria contained a plasmid for a green fluorescent protein (GFP) reporter for c-di-GMP. We used laser-scanning confocal microscopy to measure the dynamics of the GFP reporter for 3 h, beginning 1 h after introducing bacteria to the hydrogel. We controlled for the effects of changes in bacterial metabolism using a promoterless plasmid for GFP, and for the effects of light passing through different hydrogels being differently attenuated by using fluorescent plastic beads as 'standard candles' for calibration. We demonstrate that this method can measure statistically significant differences in c-di-GMP signalling associated with different PEGDA gel types and with the surface-exposed protein PilY1.

# 1. Introduction

In most natural, non-laboratory settings, bacteria live as part of biofilms, complex interacting communities often associated with surfaces [1–3]. Robust biofilms can form on a wide range of materials, are difficult to eliminate once formed, and resist both antibiotics and the immune system [4,5]. As a result, biofilms are a large and growing problem in the healthcare industry, estimated to be found in 80% of all microbial infections [6–8]. Biofilms can grow on many types of medical devices, from catheters to implants [9–12]. These complications are common; previous work has shown the infection rate of urinary catheters to be 26.6–35% and that of orthopaedic implants to be 5–40% [6]. Given the difficulty of eliminating biofilms once they are formed, the most common approach to dealing with biofilms on medical devices is to remove the device, with consequent increased risk and suffering to the patient, and increased cost to the patient and the healthcare system at large [13]. It would be better to have medical devices that resist the formation of biofilms in the first place.

To this end, coatings have been developed that hinder biofilm growth by killing bacteria (these are antimicrobial surfaces) or by resisting the attachment of bacteria (these are antifouling surfaces). The antimicrobial properties of surfaces are a result of either directly incorporating antibiotics into a coating or using a metal such as silver [12]. Antifouling coatings are made of materials, ranging from hydrogels to ceramics, that prevent attachment via specially chosen surface properties such as hydrophilicity, topology and interfacial energy [14–16]. Although antifouling coatings are effective in slowing bacterial colonization of the surface, their effectiveness is limited since they only target the attachment stage of biofilm development (in contrast, coatings can release antimicrobial agents across a range of timescales could that include biofilm maturation).

Bacteria have multiple, redundant mechanisms for attachment; moreover, conditioning of the surface, by both bacteria-produced materials and host materials, can reduce or eliminate their attachment-preventing properties over time. A hitherto-unexplored alternative is to target the biological signalling that controls biofilm development by controlling gene expression. Such an approach might be less susceptible to thwarting by redundant attachment mechanisms and surface conditioning. Furthermore, targeting the signalling necessary for biofilm development should also have the potential to be combined with antifouling or bactericidal materials, to achieve a multi-pronged approach to hindering biofilm development. However, to accomplish this, it is first necessary to establish methods that allow quantitative comparison of the effects of different biomedical materials on bacterial signalling and its dynamics.

*Pseudomonas aeruginosa* is an opportunistic pathogen that is responsible for many nosocomial infections, in large part due to its ability to form biofilms [17]. To form biofilms, planktonic *P. aeruginosa* attach to a surface and increase their intracellular concentration of the second messenger cyclic-di-GMP (c-di-GMP) [18–20]. C-di-GMP is the best-studied of the cyclic dinucleotide signalling molecules and the most widespread among bacterial species [21]. C-di-GMP coordinates both flagella- and pilus-driven motility (and can, in turn, be impacted by active motility elements such as flagellar stators [22] and a pilus motor [23]), virulence and progression through the cell cycle, as well as biofilm formation and the production and secretion of proteins and polysaccharides such as biofilm matrix materials [21,24]. C-di-GMP has also been associated with promoting antibiotic tolerance [25–27]. C-di-GMP controls the expression of many genes involved in biofilm formation; high intracellular levels of c-di-GMP are necessary for the shift from the non-biofilm, planktonic state to the biofilm state [28].

Poly(ethylene glycol) diacrylate (PEGDA) hydrogels are used in studies probing bacterial attachment and in many biomedical applications [29–32]. PEGDA hydrogels are biocompatible and have been used as three-dimensional constructs for tissue engineering [33,34] and matrices for controlled release of drugs [35,36]. In addition to tunable matrix stiffness, the transparency of PEGDA hydrogels facilitates the confocal fluorescence imaging for the quantification of adhered bacterial cells. This optical property makes PEGDA an ideal test material for a pilot study to develop a method for measuring the effects of real-world biomedical materials on c-di-GMP signalling by *P. aeruginosa*.

It is well established that *P. aeruginosa* cells increase intracellular levels of c-di-GMP after attaching to rigid solids such as glass, which has an elastic modulus of about approximately 20 GPa [23,37,38]. Very little is known about how c-di-GMP signalling is impacted by substrate mechanics [38]. It is well known that cells from higher eukaryotes sense and respond to the mechanics of their substrate [39–41]. Indeed, a large body of research has studied mechanotransduction in higher eukaryotes [42–44]. However, very little is known about prokaryotic mechanosensing in any context, including that of biofilm formation. We have shown that mechanical shear can act as a cue for surface attachment and result in increased levels of c-di-GMP [23]. More recently, others have shown that bacteria can sense and respond to the intensity of flow of the liquid medium that surrounds them [45]. This suggests the possibility that

bacteria could respond to other mechanical cues from their environment, such as the mechanics of the substrate to which they attach.

By modulating the macromer molecular weight and concentration or introducing cross-linker molecules, researchers have fabricated PEGDA hydrogels with varied cross-link densities [36,46,47]. This provides control over the hydrogel mechanical properties, such as compressive modulus. Browning *et al.* [46] reported this compositional control of PEGDA hydrogel stiffness with a compressive modulus range from 50 to 2500 kPa. Previous researchers have found that fewer bacteria adhere to soft PEGDA hydrogels than to stiff PEGDA hydrogels [29,48]. C-di-GMP upregulates the production of extracellular polysaccharides, which promote bacterial adhesion to surfaces, and downregulates motility, which promotes detachment from surfaces [49–54]. Thus, if softer PEGDA hydrogels might be associated with lower c-di-GMP signalling than are stiffer PEGDA hydrogels, this would be expected to lead to lower adhesion. Therefore, in the course of demonstrating an approach to measuring the effects of a biomedical material on c-di-GMP signalling, we will also evaluate the degree to which this signalling correlates with substrate mechanics.

In an extension of the technique we used previously [23], we use quantitative confocal microscopy to measure changes in the intensity of a green fluorescent protein (GFP) reporter for cdrA, which has previously been verified as a reporter for c-di-GMP [55]. In our previous work, the only substrates used were glass coverslips. In this work, we use different types of hydrogel substrate, and therefore we present a calibration that allows us to account for optical attenuation caused by the different types of substrate. Measurements were done every 30 min beginning 60 min after surface attachment and lasting until 240 min after surface attachment. We compare signalling timecourses for bacteria attached to a stiff PEGDA hydrogel, with elastic modulus 4000 kPa, with those for bacteria attached to a soft PEGDA hydrogel, with elastic modulus 50 kPa. The membrane protein PilY1 has been implicated as a possible mechanosensor, so we also compare signalling timecourses for wild-type (WT) and cells lacking PilY1 ($\Delta pilY1$).

We find that that *P. aeruginosa* cells have a faster, stronger c-di-GMP signalling response subsequent to the attachment on a stiff PEGDA hydrogel than they do subsequent to the attachment on a soft PEGDA hydrogel. Furthermore, the differentiation between surfaces is affected by the presence of PilY1; $\Delta pilY1$ bacteria have different c-di-GMP signalling dynamics following surface attachment than do WT.

# 2. Experimental

## 2.1. Materials

Poly(ethylene glycol) (PEG, 2 kDa ($M_n = 1917$) and 10 kDa ($M_n = 12\,157$)), acryloyl chloride, triethylamine, potassium bicarbonate, sodium sulfate and Irgacure 2959 were purchased from Sigma-Aldrich (St Louis, MO). Dichloromethane and deuterated chloroform with 0.03 vol% TMS were purchased from VWR Chemicals (Radnor, PA). Diethyl ether and Dulbecco's phosphate buffered saline were purchased from Fisher Scientific (Hampton, NH). SecureSeal Imaging Spacers were purchased from Grace Bio-Labs (Bend, OR). All reagents were used as received unless specified otherwise.

## 2.2. Bacteria

We used WT *P. aeruginosa* strain PAO1 and the mutant $\Delta pilY1$, also in the PAO1 background, in our experiments [56]. PAO1 is a widely used laboratory strain. The $\Delta pilY1$ strain does not make the envelope protein PilY1. To study intracellular c-di-GMP levels, both strains were transformed with the reporter plasmid pCdrA::*gfp*. In this plasmid, the expression of GFP is under the control of the promoter for the gene *cdrA*. This gene is transcriptionally controlled by c-di-GMP and therefore increased c-di-GMP results in an increase in fluorescence intensity [55]. Both WT and $\Delta pilY1$ PAO1 were also transformed with the plasmid pMH487 instead of the reporter plasmid. The plasmid pMH487 contains a promotorless GFP gene and thereby provides a control measurement of background metabolic activity.

## 2.3. Bacterial growth and media

We grew all bacterial strains as previously described [23]. In brief, we first streaked frozen stocks onto plates made of LB agar (5 g of yeast extract, 10 g of tryptone, 10 g of sodium chloride and 15 g of agar, all from Sigma-Aldrich, per litre of deionized water) supplemented with the antibiotic gentamicin (gentamicin sulfate, Sigma-Aldrich) at 60 µg ml$^{-1}$ for the purpose of plasmid selection. After streaking, the plates were incubated at 37°C for 20 h. Subsequently, we picked a single colony and used it to inoculate 5 ml of

LB media (5 g of yeast extract, 10 g of tryptone and 10 g of sodium chloride, all from Sigma-Aldrich, per litre of deionized water) supplemented with gentamicin at 60 μg ml$^{-1}$. The resulting culture was then shaken in an orbital shaker (Labnet Orbit 1000) operating at 235 r.p.m. for a period of 16–18 h. We diluted 40 μl of the overnight culture into 5 ml of fresh LB media and these bacteria were then used in our experiments.

## 2.4. Synthesis of poly(ethylene glycol) diacrylate

PEGDA was synthesized according to previously established protocols [46,57]. In brief, dry PEG (1 mol, 2 or 10 kDa) was dissolved in anhydrous dichloromethane (0.1 M) under nitrogen atmosphere. Triethylamine (2 mol) and acryloyl chloride (4 mol) were added to the solution successively in a dropwise manner. The PEG solution was cooled in an ice bath prior to dropwise addition. The reaction was stirred for 24 h at room temperature, then washed with 2 M potassium bicarbonate (8 mol) and dried with anhydrous sodium sulfate. The product was precipitated in cold diethyl ether, filtered and dried at atmospheric pressure for 24 h and under vacuum briefly. The synthesis was confirmed with proton nuclear magnetic resonance ($^1$H–NMR). Spectra were recorded on a Varian MR400 400 MHz spectrometer using a TMS/solvent signal as an internal reference (electronic supplementary material, figure S1). Polymers with the conversion of hydroxyl to acrylate end groups greater than 90% were used in this investigation. $^1$H–NMR (CDCl$_3$): 3.6 ppm (m, –OCH$_2$CH$_2$–), 4.3 ppm (t, –CH$_2$OCO–), 5.8 and 6.4 ppm (dd, –CH = CH$_2$), and 6.1 ppm (dd, –CH = CH$_2$).

## 2.5. Hydrogel fabrication and characterization

Hydrogels were prepared by first dissolving PEGDA in deionized water at a concentration of 10 wt% 10 kDa PEGDA or 50 wt% 2 kDa PEGDA. A photoinitiator solution (Irgacure 2959, 10 wt% in 70% ethanol) was then added at 1 vol% of the precursor solution. Imaging specimens were prepared by pipetting 4 μl of the PEGDA solution into curing moulds. The mould consists of an imaging spacer liner (Grace Bio-Labs SecureSeal Imaging Spacers) placed on a coverslip and sealed against a glass plate. Hydrogels were cross-linked by a 12 min exposure to long-wave UV light (Ultraviolet Products High-Performance UV Transilluminator, 365 nm, 4 mW cm$^{-2}$, Analytik Jena). The imaging spacers used each had a single well of diameter 13 mm and the liner, which was used as the mould for casting PEGDA gels, has a thickness of about 0.05 mm (Grace Bio-Labs, personal communication). Thus, the pre-swelling thickness of PEGDA gels used for imaging was about 0.03 mm. The adhesive spacers themselves were attached to the coverslip to enclose the gel after it was cast; these spacers have a thickness of 0.12 mm. Gels were then swollen to their equilibrium height by adding liquid medium. At the start of each imaging session, the microscope objective was first focused on the coverslip bottom and then focused on the bacteria on the top of the gel. The height difference between these positions, read off the control software, gave an approximate measurement of gel thickness. Gels ranged from 0.1 to 0.13 mm in thickness.

Specimens for swelling ratio and modulus characterization were prepared by pipetting the precursor solution between glass plates spaced 1.5 mm apart and exposed to UV light 6 min on each side. Hydrogels for imaging were soaked in Dulbecco's phosphate buffered saline for 20 h prior to imaging.

### 2.5.1. Hydrogel swelling ratio

Hydrogel specimens ($T = 1.5$ mm, $D = 8$ mm, $n = 3$) were punched with an 8 mm biopsy punch (Integra Miltex) after cross-linking and swelled in deionized water for 3 h to reach equilibrium swelling. Specimens were weighed to obtain the equilibrium swelling mass ($W_s$), then dried under vacuum at room temperature for 24 h and weighed again to determine the dry mass ($W_d$). The equilibrium volumetric swelling ratio, $Q$, was calculated from the equilibrium mass swelling ratio

$$Q = \frac{W_s}{W_d}$$

Average values with standard deviation are reported.

### 2.5.2. Gel mesh size

We estimated the mesh size of our gels using a previously described method, as follows [58]. As expected, the stiff hydrogels displayed a much lower equilibrium swelling ratio (2.44 ± 0.05 standard deviation) than the soft hydrogels (18.7 ± 0.37 standard deviation).

The average molecular weight between adjacent cross-links was then estimated from the equilibrium swelling ratio using the Peppas–Merrill model [59]

$$\frac{1}{M_c} = \frac{2}{M_n} - \frac{(\bar{v}/V_1)[\ln(1-v_{2,s}) + v_{2,s} + X_1 v_{2,s}^2]}{v_{2,r}\left[(v_{2,s}/v_{2,r}^{1/3}) - \frac{1}{2}(v_{2,s}/v_{2,r})\right]},$$

(2.1)

where $M_n$ is the average molecular weight of the hydrogel macromers, $M_c$ is the average molecular weight between two adjacent cross-links, $v_{2,s}$ is the polymer volume fraction in the swollen state, $X_1$ is the Flory–Huggins polymer–solvent interaction parameter, $\bar{v}$ is the specific volume of PEGDA in its amorphous state, $V_1$ is the molar volume of the solvent and $v_{2,r}$ is the polymer fraction in the hydrogel.

Given this information, we then calculated the mesh size of our hydrogels using the formulae

$$\xi = (\overline{r_0^2})^{1/2} v_{2,s}^{-1/3}$$

(2.2)

and

$$\overline{r_0^2} = l^2 \left[2\frac{\overline{M_c}}{M_r}\right] C_n,$$

(2.3)

where $\overline{r_0^2}$ is the root mean square end-to-end distance of the polymer in its free state, $l$ is the carbon–carbon bond length, $C_n$ is the rigidity factor of polymer and $M_r$ is the molecular weight of repeating units.

The estimated mesh size of our soft (approx. 50 kPa) hydrogel was 8.3 nm and the estimated mesh size of our stiff (approx. 4000 kPa) hydrogel was 0.9 nm.

### 2.5.3. Hydrogel compressive modulus

Hydrogel specimens ($T = 1.5$ mm, $D = 8$ mm, $n = 6$) were punched from the hydrogel slab for each tested formulation after reaching equilibrium swelling. Unconstrained compression tests were run at room temperature using a dynamic mechanical analyser (RSA 3, TA Instruments) equipped with a parallel-plate compression clamp. A dynamic strain sweep was used to determine the linear viscoelastic range for each formulation. Subsequently, a strain in the upper portion of the linear viscoelastic range was used in a constant-strain frequency sweep (0.79–79 Hz). The storage modulus was recorded at 1.25 Hz for each hydrogel and reported as the compressive modulus for each composition. Mean values with their standard errors are reported.

## 2.6. Laser-scanning confocal fluorescence microscopy

For all experiments, we used an Olympus FV1000 motorized inverted IX81 microscope suite, with instrument computer running FV10-ASW v. 4.2b software, to image attached bacteria using laser-scanning confocal microscopy. To prepare the bacteria, we first diluted 40 µl of an overnight culture into 5 ml of fresh LB media containing gentamicin. We then placed an imaging spacer (Grace Bio-Labs SecureSeal Imaging Spacers) on both the microscope slide and coverslip around the PEGDA hydrogel. Twenty-five microlitres of the bacterial dilution was inoculated onto the PEGDA hydrogel substrate on a glass coverslip and sealed to a microscope slide. The slide was then placed on the microscope stage and bacteria were allowed to adhere to the hydrogel for an hour prior to imaging. During this hour 10–15 locations containing adhered bacteria were identified for subsequent time-series imaging. Imaging was done using a 60x oil-immersion objective, a 488 nm laser with a 405/488 excitation filter and an emission filter of 505/605. For each day's worth of experiments, 10–15 sites were imaged every 30 min for a total of 3 h. This timescale was chosen because not long after 240 min post-attachment the local bacterial density can be too high to allow single-cell brightness to be confidently measured; this time-span is comparable to that covered in related prior work [23,60]. This process was repeated on three different days for each condition. To image each site a confocal z-stack was taken with a depth of 6 µm and an inter-slice size of 750 nm. At each site on each day, roughly 40–90 bacterial cells were imaged at each initial time point and roughly 80–160 bacterial cells were imaged at each final time point.

## 2.7. Accounting for fluorescence attenuation

To account for the attenuation of exciting and emitted light passing through different hydrogels, we used green fluorescent beads (Dragon Green, Bangs Laboratories, Inc.) that are similar in both their excitation

and emission spectra to GFP-expressing bacteria, thus acting as a model for how the GFP excitation and emission light is affected by passing through the hydrogel. All beads were imaged using the same laser, but different intensity, photomultiplier and image acquisition settings; these were different settings to ensure the beads were not overexposed in our images. These measurements were done on three different days, with different gel preparations on each day. The numbers of beads measured on each day were: 995, 1769 and 1407 beads on the soft gels, and 262, 1752 and 1285 beads on the stiff gels. Similar attenuation factors were measured for each day.

## 2.8. Image processing and analysis

We used the Fiji distribution of ImageJ software (v. 1.52) for image processing [61]. Each z-stack was projected to create both a maximum intensity projection and an average intensity projection on the $x$–$y$ plane. The locations of single cells were determined on the average intensity projections to exclude cells that were not attached and only present in a single frame of the z-stack [61]. The mean fluorescence intensities of individual cells were then determined using the maximum intensity projections.

## 2.9. Statistics

Data acquired from image processing and analysis were then analysed in R (v. 3.6.1) to obtain our plots and statistical significance values. We used the Kolmogorov–Smirnov (KS) significance test for comparing two distributions and used a $p$-value threshold of 0.05 to determine significance.

# 3. Results and discussion

## 3.1. Hydrogel mechanics

The mechanics of PEGDA hydrogels were tuned by changing the molecular weight of the PEGDA used and PEGDA concentration in the precursor solution [46]. Specifically, we made stiff hydrogels from 50 wt% 2 kDa PEGDA and we made soft hydrogels from 10 wt% 10 kDa PEGDA. These combinations of PEGDA molecular weight and cross-linker concentration were chosen to maximize the mechanical differences between the two gels. We determined the compressive moduli of these two hydrogels using a dynamic mechanical analyser. Representative stress–strain curves for each type of gel are shown in figure 1. The stiff hydrogel had a compressive modulus of $3600 \pm 560$ kPa, and the soft hydrogel had a compressive modulus of $44 \pm 0.375$ kPa, showing a difference of two orders of magnitude (figure 2).

## 3.2. Accounting for the effects of optical attenuation and changes in bacterial metabolism

We used *P. aeruginosa* cells containing the plasmid pCdrA: GFP, which is a verified reporter for c-di-GMP [55], to monitor the dynamics of intracellular c-di-GMP for bacteria attached to a hydrogel substrate [18,37,49,50,62]. Data were collected by imaging cells with a laser-scanning confocal microscope at 30 min intervals (figure 3a–c; electronic supplementary material figures S2 and S3).

Different hydrogel substrates could cause different attenuation both of the light used for fluorescence excitation and of the fluorescently emitted GFP light. Therefore, we measured the intensities of fluorescent plastic beads on the two types of hydrogel substrates. These fluorescent beads have both excitation and emission spectra similar to those of GFP-expressing bacteria. Thus, the beads acted as a model for how the GFP excitation and emission light was affected by passing through the hydrogel. We found that the beads were brighter when imaged on the stiff hydrogels, 50 wt% 2 kDa PEGDA, than when imaged on the soft hydrogels, 10 wt% 10 kDa PEGDA, (figure 3d). We then calculated an attenuation factor of 0.662 from the ratio of the mean value of the two populations (beads on soft hydrogels to beads on stiff hydrogels). This factor, when applied to all of the fluorescence intensity values of the beads on the 50 wt% 2 kDa PEGDA hydrogels, collapsed the average fluorescence intensity to 79.6 arb. units. This is the same average fluorescence intensity value for beads on the 10 wt% 10 kDa PEGDA hydrogels (figure 3d inset). This attenuation factor was applied to raw fluorescence data for both reporter and control strains of *P. aeruginosa*. Control strains contained the promoterless GFP plasmid pMH487.

To control for differences in the baseline metabolism of bacteria across different substrates, we then subtracted the attenuation-corrected average fluorescence intensity of control bacteria at each time point

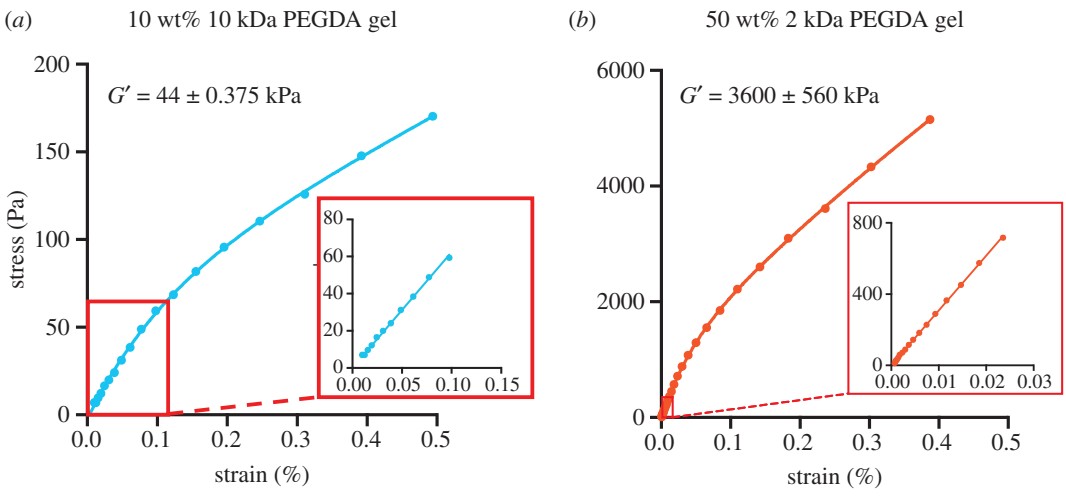

**Figure 1.** Representative stress–strain curves of (*a*) 10 wt% 10 kDa PEGDA and (*b*) 50 wt% 2 kDa PEGDA hydrogel in compression. Because the values differ by two orders of magnitude, two different *y*-axes are used. The insets display the linear viscoelastic range of the curves used for constant-strain frequency sweeps for each formulation. The average storage modulus at 1.25 Hz was 44 ± 0.375 kPa for the 10 wt% 10 kDa PEGDA gel and 3600 ± 560 kPa for the 50 wt% 2 kDa PEGDA gel. *N* = 6.

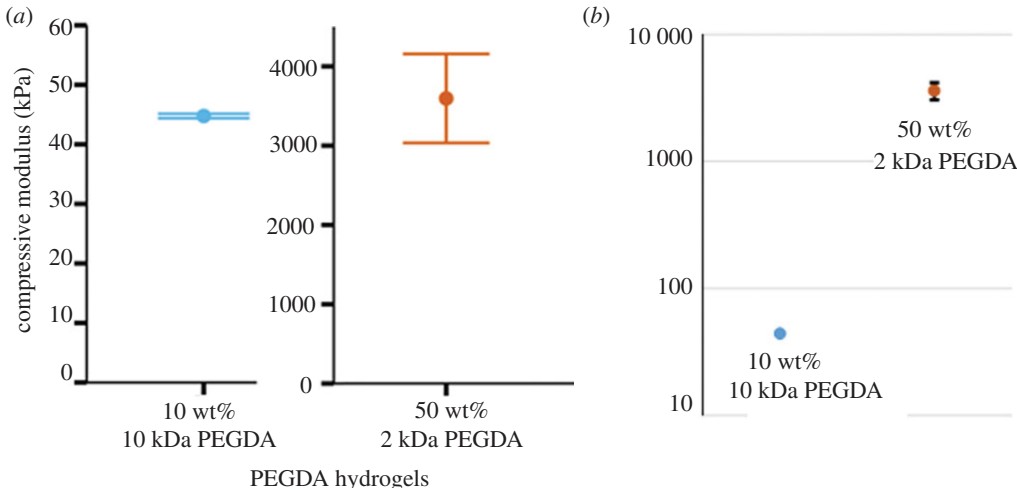

**Figure 2.** Compressive moduli of the two PEGDA hydrogel formulations used, measured using a dynamic mechanical analyser. (*a*) Because the values differ by two orders of magnitude, two different *y*-axes are used. Error bars show the standard error. (*b*) To show more clearly the relationship between the moduli of the two samples, both are plotted on the same logarithmic scale. For the softer gel, the error bars are too small to be seen on this scale. The modulus of the 10 wt% 10 kDa PEGDA gel was 44 ± 0.375 kPa and the modulus of the 50 wt% 2 kDa PEGDA gel was 3600 ± 560 kPa. *N* = 6.

from the measured intensities of reporter bacteria on the same type of gel. This removed fluorescence resulting from the metabolism of these cells and leaves only the fluorescence associated with intracellular c-di-GMP levels (figure 3*e*). This approach is rooted in that used for the original development and validation of the reporter plasmid, and its subsequent use [49,55,63,64]. We used a similar approach in our prior work on bacterial mechanosensing [23].

## 3.3. Substrates of different stiffnesses are associated with statistically significant differences in c-di-GMP signalling patterns

We used the KS significance test for comparing two distributions to compare brightness distributions of populations of bacterial cells on soft (approx. 50 kPa) and stiff (approx. 4000 kPa) hydrogels at the same time point after attachment and to compare brightness distributions on the same type of

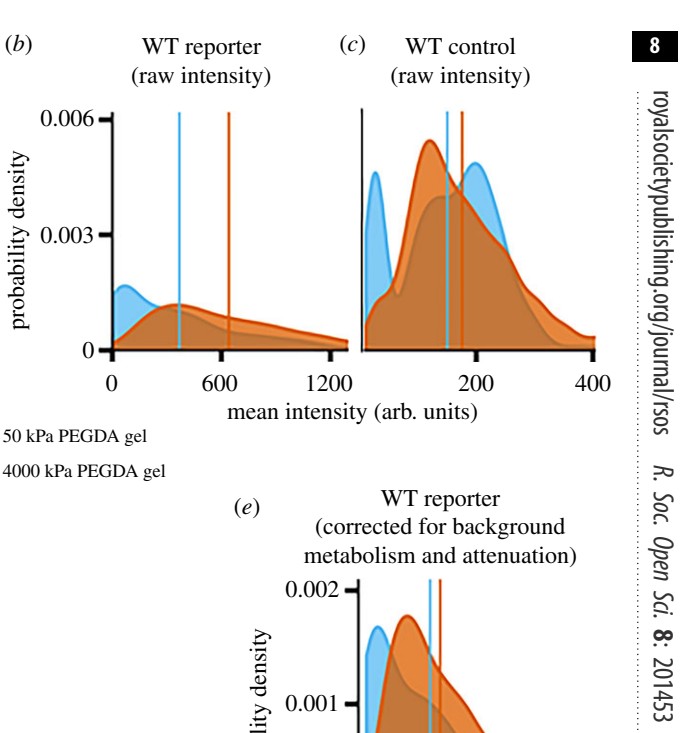

**Figure 3.** Imaging and calibration for WT PAO1. (*a*) Representative micrograph of GFP-expressing *P. aeruginosa*, attached to the soft gel substrate at the first imaging time point. A heat-map false-colour scheme is used to indicate different intensities of GFP brightness. (*b*,*c*) Distributions of measured per-cell fluorescence intensities for populations of the WT reporter (containing pCdrA::gfp) (*b*) and the WT control (containing the plasmid pMH487) (*c*). The same vertical axis is used for both (*b*) and (*c*). Distributions are shown in blue for populations on the 50 kPa PEGDA (soft) gel and in orange for populations on the 4000 kPa PEGDA (stiff) gel. The mean for each distribution is indicated with a vertical line of the appropriate colour. (*d*) The measured distributions of intensities of fluorescent beads on the soft gel, shown in blue and the stiff gel, shown in orange, are different. (*d*, inset) An attenuation factor was multiplied with the fluorescence intensities of all beads on the stiff gel, such that the mean values of the corrected intensity distributions on both gel types were equal. (*e*) Attenuation- and metabolism-corrected fluorescence intensity distributions of populations containing the reporter plasmid. Cells on the stiff gel had their measured intensities multiplied by the attenuation factor. Then, cells on both gels had the mean intensity for that gel's corresponding control population subtracted from their brightness. The average fluorescence was higher on the 4000 kPa (stiff) PEGDA gel.

hydrogel at later time points with the first-measured brightness distribution [65]. For this, we used attenuation- and metabolism-corrected brightness distributions. In every case, the *p*-value from the KS test was well below 0.01. This shows that populations on stiff and on soft hydrogel substrates had statistically significant differences throughout the 3 h after attachment to the substrate. Populations on both types of hydrogels differed in their brightness distributions from the initially measured distribution to a statistically significant level throughout the time-window of observation. From 60 to 120 min after attachment, the average brightness and the increase in fluorescence was greater for bacteria on the stiff hydrogel (figure 4*a*). However, by 150 min after attachment, bacterial populations on the soft hydrogel reached higher brightness levels than those on the stiff hydrogel (figure 4*a*).

Other researchers have previously shown that *P. aeruginosa* populations are heterogeneous in their response to surface attachment [66]. In short, they found that, upon attachment to a surface, one subpopulation of cells would robustly increase its intracellular c-di-GMP concentration and begin biofilm formation, while another subpopulation would retain low c-di-GMP levels and engage in more surface motility. To examine how heterogeneous response might interplay with substrate stiffness, we measured the skewness and kurtosis of all attenuation- and metabolism-corrected brightness

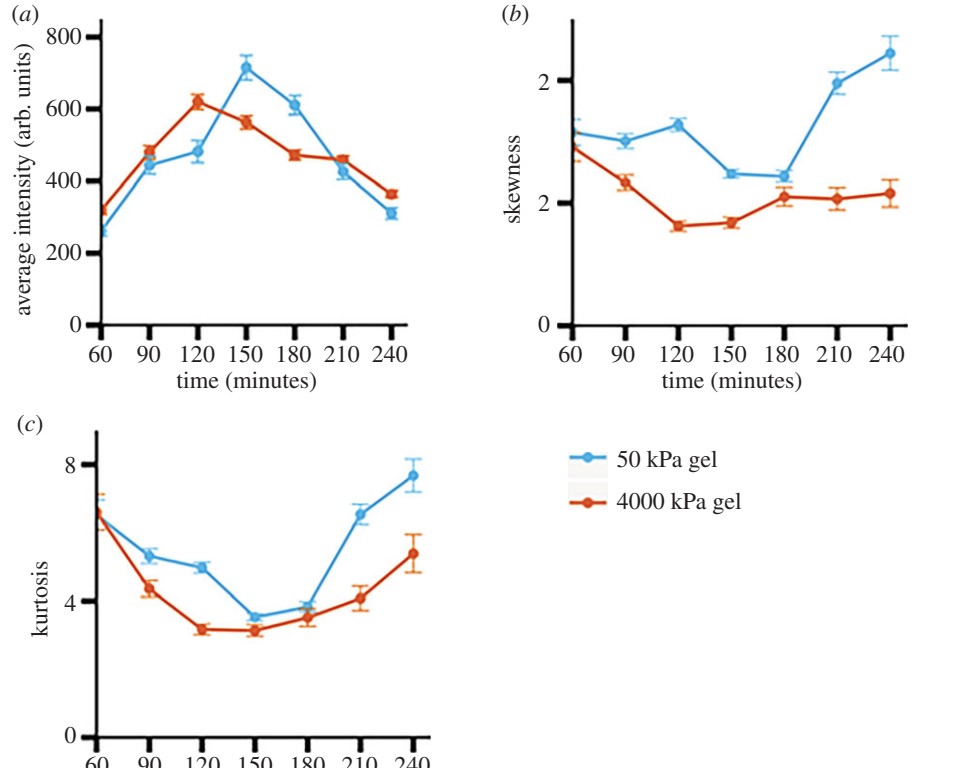

**Figure 4.** Comparison between distributions of fluorescence intensities of PA01 WT cells on the 4000 kPa (stiff) PEGDA gel and the 50 kPa (soft) PEGDA gel. *p*-values are obtained from the Kolmogorov–Smirnov test. (*a*) Average fluorescence intensities of all single cells attached to both the soft and stiff gel for 3 h. We see that the bacteria attached to the stiff gel reach their peak fluorescence earlier at 120 min in comparison to 150 min for bacteria attached to the soft gel. (*b*) Skewness of fluorescence intensity distributions at every time point. In both conditions, we see distributions that are skewed right. (*c*) Kurtosis of fluorescence intensity distributions at every time point. The kurtosis value is higher on the soft gel than on the stiff gel. This signifies a higher presence of extreme data points or outliers. Each data point shows the mean value of three independent biological replicates ($N = 3$), with error bars the standard error of the mean.

distributions—skewness to determine asymmetry and kurtosis to determine the preponderance of outliers [67]. The skewness values we measured were all positive, in agreement with our observation that measured distributions had a 'tail' on the right (brighter) side of the mean; differences in the size of the skewness measure differences in how much of the distributions is found on the right (brighter) side of the mean. Kurtosis measures how much a distribution lies in the tail(s), and high kurtosis values correspond to 'heavy' tails, or having more of the distribution farther from the mean. We found that, for all time points, the skewness and kurtosis were higher for populations attached to the softer hydrogel than for those attached to the stiffer hydrogel (figure 4*b,c*). This showed that attachment to the softer gel was associated with more outliers and therefore, a more distinct subpopulation of 'strong responders' as measured by the brightness of the GFP reporter which we use as our proxy measure for c-di-GMP concentration. This does not indicate that the population on the soft gel has a stronger overall response to surface attachment, as shown by the average values in figure 4*a*; rather, it indicates that a smaller fraction of the population responds strongly to attaching to a soft gel than responds strongly to attaching to a stiff gel. By contrast, attachment to the stiff gel was associated with fewer 'strong responder' outliers and therefore a higher proportion of the population responding with increased c-di-GMP concentration. These results suggested that substrate mechanics might impact the development of heterogeneity in populations of surface-attached bacteria.

## 3.4. Loss of the membrane protein PilY1 impacts the timescale of surface sensing

We and others have found that the envelope protein PilY1 is required for *P. aeruginosa* to increase c-di-GMP levels following attachment to a rigid surface and also strongly impacts other surface-associated

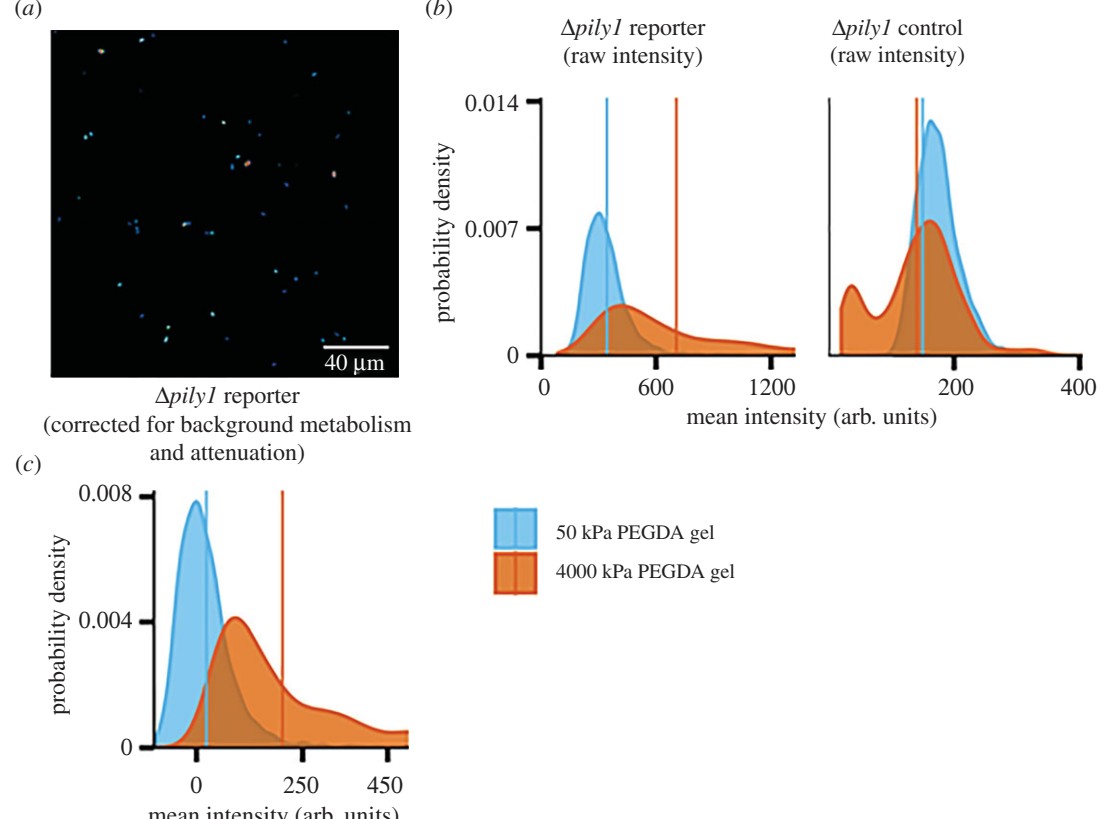

**Figure 5.** (a) Example image of attached Δ*pily1* cells, on the soft gel at the first imaging time point. (b) Uncalibrated fluorescence intensity of individual PA01 Δpily1 pCdrA::gfp cells (left) and PA01 Δ*pily1* pMH487 cells (right) on both the 10 k PEGDA gel (soft) and 2 k PEGDA gel (stiff). (c) Fluorescence intensity distributions of PA01 Δ*pily1* pCdrA::gfp cells with calibration from the fluorescent beads attenuation factor and subtracted mean intensity values from the control vector. The average fluorescence was higher on the 4000 kPa (stiff) PEGDA gel.

behaviours; these findings suggest that PilY1 could act as a mechanosensor to transduce the mechanical signal(s) resulting from surface attachment [23,37,60,62,68]. Here, we used the bacterial strain Δ*pily1*, which does not make PilY1, to demonstrate the utility of our method for probing the importance of this envelope protein for the differential response to the two PEGDA hydrogel stiffnesses, using the same experimental and analytical procedure described above for WT (figure 5; electronic supplementary material, figures S4 and S5).

Similarly to the case for the WT cells, the KS test comparing brightness distributions of populations of reporter Δ*pily1* bacteria on the two hydrogel types with each other gave *p*-values well below 0.01 at every time point. This indicates that there were statistically significant differences between the populations on the two gels at every time point. Also similarly to the case for the WT, the KS test comparing the brightness distributions of Δ*pilY1* populations at 60 min to those at subsequent time points gave *p*-values well below 0.01 in all cases but one. This indicates that there were statistically significant changes in the populations' brightness distributions after the 60 min measurement. The one exception was for bacteria attached to the stiff hydrogel at 240 min—the KS test comparing this to the 60 min case gave a *p*-value of slightly less than 0.4, indicating no statistically significant difference. Similar to our findings for WT, we found that the average brightness of Δ*pilY1* bacteria on the stiffer hydrogel was higher than that on the softer hydrogel, with statistical significance at every time point (figure 6*a*). However, the timescale of c-di-GMP increase was different in WT and Δ*pilY1* populations. WT increased brightness on the stiff hydrogel over the initial 60 min of observation and peaked at 120 min after attachment (figure 4*a*). By contrast, Δ*pilY1* cells on the stiff hydrogel did not increase their fluorescence in the initial 30 min of observation and peaked at 150 min after attachment (figure 6*a*). However, the 30 min differences were also the time-resolution of our measurements, cautioning against over-interpretation of this finding.

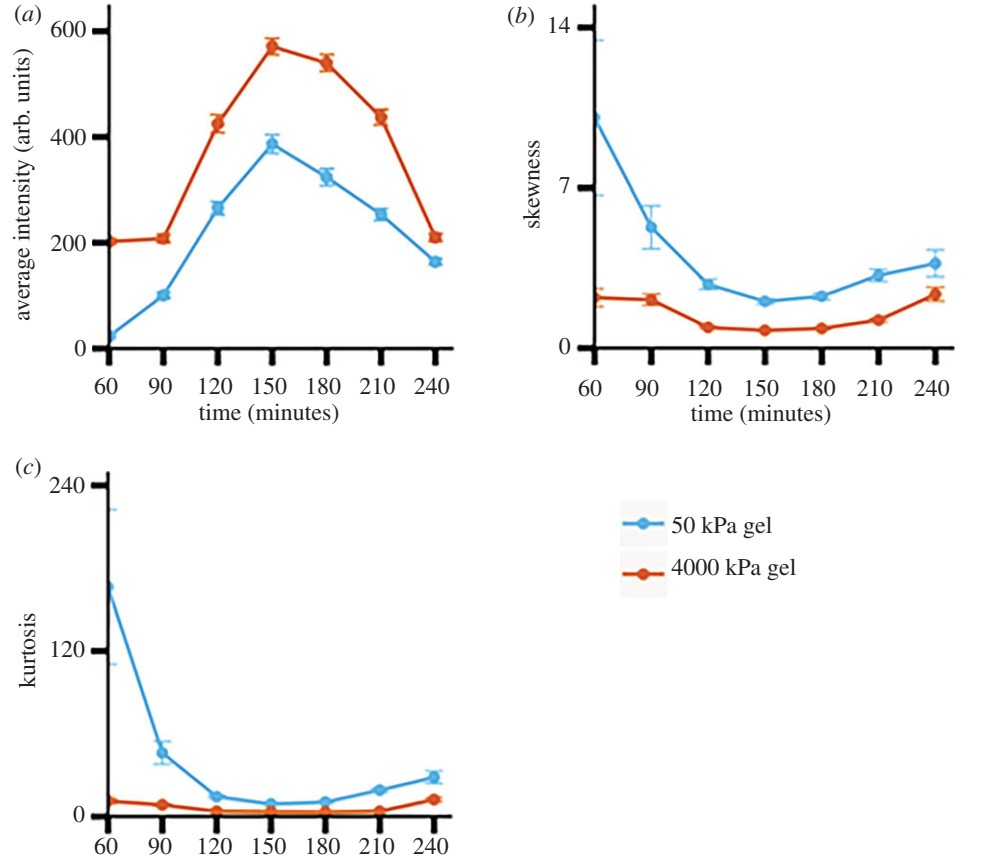

**Figure 6.** (*a*) Average fluorescence intensities of all *Δpily1* single cells attached to both the soft and stiff gel for 3 h. (*b*) Skewness of fluorescence intensity distributions at every time point. In both conditions, we see distributions that are skewed right and that the distributions from the soft gel condition are more skewed to the right. (*c*) Kurtosis of fluorescence intensity distributions at every time point. The kurtosis value is higher for the distributions of bacteria attached to soft gels at every time point. This signifies a higher presence of extreme data points or outliers. In both conditions, the kurtosis values are highest at the end time points. Each data point shows the mean value of three independent biological replicates (*N* = 3), with error bars the standard error of the mean.

## 3.5. Loss of the membrane protein PilY1 increases heterogeneity in c-di-GMP signalling on both soft and stiff substrates

In *Δpily1* populations, as for WT, the kurtosis and skewness of the brightness distributions is highest in cells attached to the soft hydrogel (figure 6*b,c*). Both kurtosis and skewness are higher for *ΔpilY1* populations at both initial and final time points (figure 6*b,c*) than for WT at the same times (figure 4*b,c*); differences are shown in electronic supplementary material, figure S6. These data demonstrate that the c-di-GMP response of *Δpily1* to surface attachment is more heavily impacted by outliers consisting of 'strong responders' than is the WT response. This result aligns with our and others' previous findings that PilY1 is used to regulate the response of *P. aeruginosa* to mechanical inputs and that PilY1 is important for signalling associated with surface sensing, including but not limited to c-di-GMP signalling [23,37,60,69].

## 3.6. Potential role of gel mesh size

However, it is possible that the *ΔpilY1* bacterial cells are differentiating between our two substrates not based on mechanical cues, but on another feature of our substrates. We speculate the mesh size of our two substrates could be affecting our measurements of surface response in addition to their stiffnesses. We believe this could be affecting out results as a higher mesh size would allow faster diffusion of nutrients and therefore a more nutrient-rich environment.

To address this, we estimated the mesh size of our gels as described in the Experimental section.

The estimated mesh size of our soft (approx. 50 kPa) hydrogel was 8.3 nm and the estimated mesh size of our stiff (approx. 4000 kPa) hydrogel was 0.9 nm, both well below the approximately 1 μm bacterial size. These are comparable to the mesh sizes previously measured for comparable PEGDA gels [46,47,70]. Given this information, we then sought to understand how a different mesh size will impact the diffusion of particles and nutrients through the gel. Work done by others has shown that in PEG hydrogels as the mesh size decreases so does the diffusion of particles. Specifically, diffusion and mesh size are related by the equation below [71].

$$\frac{D_s}{D_0} = \exp\left(-\frac{\pi(r_h + r_f)^2}{(\xi + 2r_f)^2}\right), \tag{3.1}$$

where $D_0$ is the diffusion coefficient of the particular solute in PBS alone, $D_s$ is the diffusion coefficient, $r_h$ is the solute hydrodynamic radius, $r_f$ is the polymer fibre radius and $\xi$ is the mesh size of the hydrogel. From this equation, we can see that as mesh size increase so does the diffusivity of a given material. Therefore, given that both hydrogels are made with similar polymers, we expect that diffusion of nutrients and growth substrates would be faster through the softer hydrogel with a larger mesh size than through the stiffer hydrogel. Thus, bacteria on the softer hydrogel could have access to a larger effective volume of growth substrate than will bacteria on the stiffer hydrogel, which could impact both metabolism and c-di-GMP signalling. Furthermore, the impact of mesh size on access to growth substrate and therefore on metabolism might be a reason that WT bacteria become brighter on the soft gel than on the hard gel between approximately 140 and approximately 200 min (figure 4a). We will investigate the effect of mesh size on bacterial response to attachment further in later work.

## 3.7. Possible effects of gel heterogeneity

Figure 3d shows brightness distributions for fluorescent beads that are both asymmetric and do not perfectly collapse onto each other when scaled. The causes for these phenomena, which may suggest a limit to the method we present here, are not known. The asymmetric distribution probably corresponds to an asymmetric distribution of bead brightnesses, and perhaps also is impacted by lensing or attenuation artefacts caused by internal heterogeneities in the gel substrates. The differences in the distribution shapes, shown by the collapse not resulting in a perfect match, probably arise from such heterogeneity-caused optical artefacts. The asymmetries and shape differences seen for beads are much smaller than the ones we measure for bacterial populations on the same gels, so the overall conclusions of this work and the method shown here are not affected. However, future work using higher-resolution, more fine-grained signalling measurements may need to account for internal heterogeneities in gel substrates.

# 4. Summary and conclusion

We have demonstrated that quantitative confocal microscopy and image analysis, combined with calibration for both optical attenuation and bacterial metabolism, can measure statistically significant differences in the production of CdrA, which can act as a proxy measure for c-di-GMP. As proof-of-concept for using this technique on real pathogens attached to a real-world biomedical material, we have found that *P. aeruginosa* WT PAO1 responded more quickly, and with more strong responders, when they attached to a stiff PEGDA hydrogel surface than when they attach to a soft PEGDA hydrogel surface. C-di-GMP is a second messenger that controls the transition from a planktonic to a biofilm state, so this may indicate that we should expect different dynamics of biofilm development on gels with different mechanics. We will examine this in future work.

Finally, we saw that the discriminatory response to PEGDA hydrogels of different compressive modulus, by different levels and timescales of c-di-GMP upregulation, was altered when the envelope protein PilY1 was lost. This suggested a role for this protein in the preferential rapid response to a stiffer surface. This is plausible given previous work showing that PilY1 contains a domain homologous to the von Willenbrand factor, a domain present in eukaryotic mechanosensing proteins [62].

Data accessibility. Data are provided as electronic supplementary material.
Authors' contributions. J.B. carried out the bacterial work and data analysis, participated in gel preparation and characterization and drafted and revised the manuscript; Z.L. prepared and characterized gels and their

components and critically revised the manuscript; E.M.C.-H. conceived of the study, designed the study, coordinated the hydrogel portion of the work and critically revised the manuscript; V.D.G. conceived of the study, designed the study, coordinated the bacterial portion of the work and helped draft and revise the manuscript. All authors gave final approval for publication and agree to be held accountable for the work performed therein.

Competing interests. We declare we have no competing interests.

Acknowledgements. We thank Liyun Wang (University of Texas at Austin) for discussions on imaging bacterial cells. We thank Taneidra Buie (University of Texas at Austin) for providing assistance while working with PEGDA gels. Zilei (Eva) Chen and Mara Eccles performed image analysis to count the bacteria on gels. This work was supported by grants from the National Science Foundation (grant no. 1727544) and the National Institutes of Health, Institute of Allergies and Infectious Disease (grant no. R01-AI121500), both to V.D.G.

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
