## [Reviewer comments · Royal Society Open Science]

Review History

RSOS-201453.R0 (Original submission)

Review form: Reviewer 1

Is the manuscript scientifically sound in its present form?

Yes

Are the interpretations and conclusions justified by the results?

Yes

Is the language acceptable?

Yes

Do you have any ethical concerns with this paper?

Yes

Have you any concerns about statistical analyses in this paper?

No

Recommendation?

Major revision is needed (please make suggestions in comments)

Comments to the Author(s)

General:

Wording in the introduction could be improved to improve readability and provide more details. Overall descriptions of the methodology seem very thorough but the authors should consider moving mesh size calculations to the methods. It seems that the only time period assessed in the paper was three hours which is fine for assessing initial adhesion but doesn't necessarily cover any potential changes over a longer timescale; the justification for selecting 3 hours should be noted. Overall, presentation of the results is nice but could use a few minor improvements for either clarification or readability. In both presentation of results and the discussion of those results, there is a heavy statistical overtone. Clarification of some statistical terms would be appreciated and perhaps the usage of some of those statistics to reach conclusions is a bit overreaching. However, to the credit of the authors, they acknowledge this to some extent in certain sections of the paper.

Specific:

- P3, Line 21-22- Technically speaking biofilms aren't responsible for infection, maybe rephrase to "biofilms are found in" or "biofilms comprise of"
- P3, Line 46-48- Maybe here it might be worth mentioning the coatings that have been designed to release antibacterial agents/etc. While they may technically address pre-attachment and attachment itself, they can technically still be released during (or after) maturation.
- P4, Line 22- Is c-di-GMP is important for biological things besides biofilm formation. Could the selection and relevance of c-di-GMP be further explained?
- P4, Line 31- Please provide a better justification/explanation for why PEGDA is an ideal material. The reviewer assumes because it is transparent but this (or other rationales) should be stated
 - Why was DMA used to measure the compressive modulus? Was rheology used as a comparison? How does the compressive modulus compare to an Young's modulus? Could you please put your mechanical data into context with published literature.
- P6, Line 20- Could the authors please clarify if the PAO1 pseudomonas is that a clinical strain or more of a "lab strain"
- P9 Line 53-54- Why were two different PEG precursors used in this work? Could this be further clarified please. What was the effect on hydrogel mesh size, crosslinking ratio, surface/bulk chemistry. It is unclear why the same MW was not used to minimize potential differences between gels. The reviewer notes that the authors discuss mesh size later on in the manuscript, but it would be helpful to introduce this concept earlier.
 - It is unclear why the methods for calculating mesh size aren't provided in the methods section. There is no need to detail this in the results section.
 - The authors should compare their calculated mesh size to that in literature.
 - Figure 1 would be improved if both samples were provided on the same axis scale (log scale perhaps) to show the relationship between the two samples.
 - Figure 2- Part b could be improved via a better caption to clarify the difference between WT reporter and WT Control and why they are on opposite sides of the graph.
 - Figure 3- An explanation of kurtosis could prove useful to readers that aren't as versed in the jargon of statistics.
- P14 Line 29-30- "To examine how heterogeneous response might interplay with substrate stiffness, we measured the skewness and kurtosis of all attenuation- and metabolism-corrected brightness distributions - skewness to determine asymmetry and kurtosis to determine the preponderance of outliers (58). We found that, for all timepoints, the skewness and kurtosis were higher for populations attached to the softer hydrogel than for those attached to the stiffer hydrogel (Fig. 3 B and C). This showed that attachment to the softer gel was associated with more outliers and therefore, a more distinct subpopulation of "strong responders". This statement is hard to understand/justify. The reviewer feels like a definitive conclusion to this issue can only be reached by looking at the genetics and not statistics alone. Could the authors please explain this further?

Review form: Reviewer 2

Is the manuscript scientifically sound in its present form?

Yes

Are the interpretations and conclusions justified by the results?

Yes

Is the language acceptable?

Yes

Do you have any ethical concerns with this paper?

No

Have you any concerns about statistical analyses in this paper?

No

Recommendation?

Accept with minor revision (please list in comments)

Comments to the Author(s)

In this manuscript, Blacutt et al. quantitatively characterized mechanobiological responses of *P. aeruginosa* on the stiffness of substrates for biofilm formation, using PEGDA hydrogels of higher and lower stiffness and confocal imaging. For rigorous fluorescent imaging-based evaluation, they calibrated attenuation of emitted fluorescent light through hydrogels and baseline metabolism of the tested cells. The authors' results clearly show differences in the fluorescence signal between the hard and soft hydrogel cases. They repeated their experiment using the bacteria strain not producing the envelope protein, and discussed a possibility for the bacteria cell could sense differences in pore size in addition to stiffness. The manuscript is well organized, and results are well presented. The conclusion is well supported by the presented data. I suggest minor revision with the following questions and suggestions.

1. Line 39-41 in Page 1: It would be more rigorous to indicate that E values are approximate values by using \sim .
2. What is the gel thickness for bacteria culture? The thickness of hydrogel samples for swelling and stiffness measurement is shown clearly, but the gel thickness for bacteria culture is not easily found. It would be informative to clearly show the gel thickness of this case, because cells are known to sense the stiffness of the substrate or container below the gel (i.e., effective stiffness of hydrogel).
3. For the compressive modulus measurement part, it is recommended to include examples of the measured stress-strain curves in the SI or in Figure 1 (as insets) to indicate the strain range used for the modulus measurement.
4. One hour was allowed for bacteria to adhere to hydrogel, and I wonder how this time duration was determined. Could imaging be done 0.5 hour after the inoculation of hydrogels?
5. The last line in Page 8: How many cells were typically imaged at each time point? Since probability distribution curves are shown in the manuscript, providing this number (rough estimate or range) seems required for supporting the reliability of the curves.

6. Figures 2A & 4A: It would be informative to provide the time point and gel type (hard or soft) for the shown image.
7. Figure 2C: I just wonder why the curves are not symmetric and why they do not collapse even after correction. Are these due to any possible limitations of the authors' method? Then, don't they have to be considered in analyzing images of bacteria?
8. How was the attenuation factor of 0.662 shown in Page 12 determined? Is it just a ratio of the mean values?
9. Figure 3A: The brightness level of the soft gel case became higher than that of the hard gel between ~140 min and 200 min. What does this mean?
10. Line 38-45 in Page 14: What are "strong responders" exactly? Are they related to increased c-di-GMP concentration ("strong" suggests this correlation)? If so, are more strong responders on the soft gel contradict to the higher population of bacteria with increased c-di-GMP concentration? I felt this part is confusing.
11. Repeatability: Figures 3 and 5 show error bars, and I wonder if these error bars are standard deviation from repeated experiments. The bottom of Page 8 shows that the whole process of imaging was repeated on three different days. Does this mean that the n values for the graphs in Figures 3 and 5 are 3? I believe it is very important to clearly explain how repeatable the authors' experiment was.

In this sense, I wonder whether the attenuation factor shown in Page 12 was measured repeatedly and whether the authors got similar values as 0.662. This is also important because similar attenuation factor values obtained from repeated experiments would show that the authors' method is repeatable.

Also, I wonder how repeatable the average fluorescence intensity of control bacteria was. If this were not repeatable (similar between repeated experiments), the authors' way of removing the baseline metabolism level could be problematic.

Decision letter (RSOS-201453.R0)

Dear Dr Gordon

On behalf of the Editors, we are pleased to inform you that your Manuscript RSOS-201453 "Quantitative confocal microscopy and calibration for measuring differences in cyclic-di-GMP signaling by bacteria on biomedical hydrogels" has been accepted for publication in Royal Society Open Science subject to minor revision in accordance with the referees' reports. Please find the referees' comments along with any feedback from the Editors below my signature.

Please submit your revised manuscript and required files (see below) no later than 7 days from today's (ie 19-Oct-2020) date. Note: the ScholarOne system will 'lock' if submission of the revision is attempted 7 or more days after the deadline. If you do not think you will be able to meet this deadline please contact the editorial office immediately.

on behalf of Dr Sean Murray (Associate Editor) and Pietro Cicuta (Subject Editor)
openscience@royalsociety.org

Reviewer comments to Author:

Reviewer: 1

Comments to the Author(s)

General:

Wording in the introduction could be improved to improve readability and provide more details. Overall descriptions of the methodology seem very thorough but the authors should consider moving mesh size calculations to the methods. It seems that the only time period assessed in the paper was three hours which is fine for assessing initial adhesion but doesn't necessarily cover any potential changes over a longer timescale; the justification for selecting 3 hours should be noted. Overall, presentation of the results is nice but could use a few minor improvements for either clarification or readability. In both presentation of results and the discussion of those results, there is a heavy statistical overtone. Clarification of some statistical terms would be appreciated and perhaps the usage of some of those statistics to reach conclusions is a bit overreaching. However, to the credit of the authors, they acknowledge this to some extent in certain sections of the paper.

Specific:

- P3, Line 21-22- Technically speaking biofilms aren't responsible for infection, maybe rephrase to "biofilms are found in" or "biofilms comprise of"
- P3, Line 46-48- Maybe here it might be worth mentioning the coatings that have been designed to release antibacterial agents/etc. While they may technically address pre-attachment and attachment itself, they can technically still be released during (or after) maturation.
- P4, Line 22- Is c-di-GMP is important for biological things besides biofilm formation. Could the selection and relevance of c-di-GMP be further explained?
- P4, Line 31- Please provide a better justification/explanation for why PEGDA is an ideal material. The reviewer assumes because it is transparent but this (or other rationales) should be stated
- Why was DMA used to measure the compressive modulus? Was rheology used as a comparison? How does the compressive modulus compare to an Young's modulus? Could you please put your mechanical data into context with published literature.

- P6, Line 20- Could the authors please clarify if the PAO1 pseudomonas is that a clinical strain or more of a “lab strain”
- P9 Line 53-54- Why were two different PEG precursors used in this work? Could this be further clarified please. What was the effect on hydrogel mesh size, crosslinking ratio, surface/bulk chemistry. It is unclear why the same MW was not used to minimize potential differences between gels. The reviewer notes that the authors discuss mesh size later on in the manuscript, but it would be helpful to introduce this concept earlier.
- It is unclear why the methods for calculating mesh size aren't provided in the methods section. There is no need to detail this in the results section.
- The authors should compare their calculated mesh size to that in literature.
- Figure 1 would be improved if both samples were provided on the same axis scale (log scale perhaps) to show the relationship between the two samples.
- Figure 2- Part b could be improved via a better caption to clarify the difference between WT reporter and WT Control and why they are on opposite sides of the graph.
- Figure 3- An explanation of kurtosis could prove useful to readers that aren't as versed in the jargon of statistics.
- P14 Line 29-30- “To examine how heterogeneous response might interplay with substrate stiffness, we measured the skewness and kurtosis of all attenuation- and metabolism-corrected brightness distributions - skewness to determine asymmetry and kurtosis to determine the preponderance of outliers (58). We found that, for all timepoints, the skewness and kurtosis were higher for populations attached to the softer hydrogel than for those attached to the stiffer hydrogel (Fig. 3 B and C). This showed that attachment to the softer gel was associated with more outliers and therefore, a more distinct subpopulation of “strong responders”. This statement is hard to understand/justify. The reviewer feels like a definitive conclusion to this issue can only be reached by looking at the genetics and not statistics alone. Could the authors please explain this further?

Reviewer: 2

Comments to the Author(s)

In this manuscript, Blacutt et al. quantitatively characterized mechanobiological responses of *P. aeruginosa* on the stiffness of substrates for biofilm formation, using PEGDA hydrogels of higher and lower stiffness and confocal imaging. For rigorous fluorescent imaging-based evaluation, they calibrated attenuation of emitted fluorescent light through hydrogels and baseline metabolism of the tested cells. The authors' results clearly show differences in the fluorescence signal between the hard and soft hydrogel cases. They repeated their experiment using the bacteria strain not producing the envelope protein, and discussed a possibility for the bacteria cell could sense differences in pore size in addition to stiffness. The manuscript is well organized, and results are well presented. The conclusion is well supported by the presented data. I suggest minor revision with the following questions and suggestions.

1. Line 39-41 in Page 1: It would be more rigorous to indicate that E values are approximate values by using ~.
2. What is the gel thickness for bacteria culture? The thickness of hydrogel samples for swelling and stiffness measurement is shown clearly, but the gel thickness for bacteria culture is not easily found. It would be informative to clearly show the gel thickness of this case, because cells are known to sense the stiffness of the substrate or container below the gel (i.e., effective stiffness of hydrogel).
3. For the compressive modulus measurement part, it is recommended to include examples of the measured stress-strain curves in the SI or in Figure 1 (as insets) to indicate the strain range used for the modulus measurement.

4. One hour was allowed for bacteria to adhere to hydrogel, and I wonder how this time duration was determined. Could imaging be done 0.5 hour after the inoculation of hydrogels?

5. The last line in Page 8: How many cells were typically imaged at each time point? Since probability distribution curves are shown in the manuscript, providing this number (rough estimate or range) seems required for supporting the reliability of the curves.

6. Figures 2A & 4A: It would be informative to provide the time point and gel type (hard or soft) for the shown image.

7. Figure 2C: I just wonder why the curves are not symmetric and why they do not collapse even after correction. Are these due to any possible limitations of the authors' method? Then, don't they have to be considered in analyzing images of bacteria?

8. How was the attenuation factor of 0.662 shown in Page 12 determined? Is it just a ratio of the mean values?

9. Figure 3A: The brightness level of the soft gel case became higher than that of the hard gel between ~140 min and 200 min. What does this mean?

10. Line 38-45 in Page 14: What are "strong responders" exactly? Are they related to increased c-di-GMP concentration ("strong" suggests this correlation)? If so, is more strong responders on the soft gel contradict to the higher population of bacteria with increased c-di-GMP concentration? I felt this part is confusing.

11. Repeatability: Figures 3 and 5 show error bars, and I wonder if these error bars are standard deviation from repeated experiments. The bottom of Page 8 shows that the whole process of imaging was repeated on three different days. Does this mean that the n values for the graphs in Figures 3 and 5 are 3? I believe it is very important to clearly explain how repeatable the authors' experiment was.

In this sense, I wonder whether the attenuation factor shown in Page 12 was measured repeatedly and whether the authors got similar values as 0.662. This is also important because similar attenuation factor values obtained from repeated experiments would show that the authors' method is repeatable.

Also, I wonder how repeatable the average fluorescence intensity of control bacteria was. If this were not repeatable (similar between repeated experiments), the authors' way of removing the baseline metabolism level could be problematic.

===PREPARING YOUR MANUSCRIPT===

===PREPARING YOUR REVISION IN SCHOLARONE===

<https://royalsociety.org/journals/authors/author-guidelines/#data>. You should ensure that you cite the dataset in your reference list. If you have deposited data etc in the Dryad repository,

please only include the 'For publication' link at this stage. You should remove the 'For review' link.

Author's Response to Decision Letter for (RSOS-201453.R0)

See Appendix A.

Decision letter (RSOS-201453.R1)

Dear Dr Gordon,

It is a pleasure to accept your manuscript entitled "Quantitative confocal microscopy and calibration for measuring differences in cyclic-di-GMP signaling by bacteria on biomedical hydrogels" in its current form for publication in Royal Society Open Science.

Kind regards,
Royal Society Open Science Editorial Office
Royal Society Open Science

on behalf of Dr Sean Murray (Associate Editor) and Pietro Cicuta (Subject Editor)
openscience@royalsociety.org

Appendix A

We thank the editor and both reviewers for their time and attention to our paper. In response to their comments, we have made extensive revisions, detailed below.

Reviewer: 1

Comments to the Author(s)

General:

Wording in the introduction could be improved to improve readability and provide more details.

Response: Thank you.

Changes: We have made changes to the text of the Introduction, indicated by highlighting in the revised manuscript. We have also broken long paragraphs up into shorter paragraphs, to improve readability.

Overall descriptions of the methodology seem very thorough but the authors should consider moving mesh size calculations to the methods.

Response: Thank you and done.

Changes: As requested, we have moved the mesh size calculations to the Methods section.

It seems that the only time period assessed in the paper was three hours which is fine for assessing initial adhesion but doesn't necessarily cover any potential changes over a longer timescale; the justification for selecting 3 hours should be noted.

Response: Thank you and done. We have clarified the justification for this timescale by adding the highlighted statement below.

Changes: "This timescale was chosen because not long after 240 minutes post-attachment the local bacterial density can be too high to allow single-cell brightness to be confidently measured; this time-span is comparable to that covered in related prior work (1, 2)." Here, the reference numbers correspond to references at the end of this Response. Different reference numbers are found in the manuscript.

Overall, presentation of the results is nice but could use a few minor improvements for either clarification or readability. In both presentation of results and the discussion of those results, there is a heavy statistical overtone. Clarification of some statistical terms would be

appreciated and perhaps the usage of some of those statistics to reach conclusions is a bit overreaching. However, to the credit of the authors, they acknowledge this to some extent in certain sections of the paper.

Response: Thank you and done. We have added a further brief explanation of skewness and kurtosis in the main manuscript text, before the callout to Figure 4 B and C.

Changes: “The skewness values we measured were all positive, in agreement with our observation that measured distributions had a “tail” on the right (brighter) side of the mean; differences in the size of the skewness measure differences in how much of the distributions is found on the right (brighter) side of the mean. Kurtosis measures how much a distribution lies in the tail(s) and high kurtosis values corresponds to “heavy” tails, or having more of the distribution farther from the mean.”

Specific:

• P3, Line 21-22- *Technically speaking biofilms aren't responsible for infection, maybe rephrase to “biofilms are found in” or “biofilms comprise of”*

Response: Thank you and done. We have rephrased this.

Changes: “As a result, biofilms are a large and growing problem in the healthcare industry, estimated to be found in 80% of all microbial infections”

• P3, Line 46-48- *Maybe here it might be worth mentioning the coatings that have been designed to release antibacterial agents/etc. While they may technically address pre-attachment and attachment itself, they can technically still be released during (or after) maturation.*

Response: Thank you and done. We have added a parenthetical note to this effect.

Changes: “Although antifouling coatings are effective in slowing bacterial colonization of the surface, their effectiveness is limited since they only target the attachment stage of biofilm development (in contrast, coatings can release antimicrobial agents across a range of timescales could that include biofilm maturation).”

• P4, Line 22- *Is c-di-GMP is important for biological things besides biofilm formation. Could the selection and relevance of c-di-GMP be further explained?*

Response: Thank you and done. We have added the following text and references (numbering is for the references at the end of this Response; different numbers are found in the manuscript).

Changes: “C-di-GMP is the best-studied of the cyclic dinucleotide signaling molecules and the most widespread among bacterial species (3). C-di-GMP coordinates both flagella- and pilus-driven motility (and can, in turn, be impacted by active motility elements such as flagellar stators (4) and a pilus motor (1)), virulence, and progression through the cell cycle, as well as biofilm formation and the production and secretion of proteins and polysaccharides such as biofilm matrix materials (3, 5). C-di-GMP has also been associated with promoting antibiotic tolerance (6-8).”

• *P4, Line 31- Please provide a better justification/explanation for why PEGDA is an ideal material. The reviewer assumes because it is transparent but this (or other rationales) should be stated*

Response: Thank you for your feedback. In addition to the tunable mechanical properties and general biocompatibility, the transparency of PEGDA facilitated the fluorescence imaging of bacterial cells. We have added this rationale to the revised manuscript with mechanical tunability discussed in the next paragraph in detail (numbering is for the references at the end of this Response; different numbers are found in the manuscript).

Changes: “PEGDA hydrogels are biocompatible and have been used as 3D constructs for tissue engineering (9, 10) and matrices for controlled release of drugs (11, 12). In addition to tunable matrix stiffness, the transparency of PEGDA hydrogels facilitates the confocal fluorescence imaging for the quantification of adhered bacterial cells. This makes PEGDA an ideal test material for a pilot study to develop a method for measuring the effects of real-world biomedical materials on c-di-GMP signaling by *P. aeruginosa*.”

• *Why was DMA used to measure the compressive modulus? Was rheology used as a comparison? How does the compressive modulus compare to an Young's modulus? Could you please put your mechanical data into context with published literature.*

Response: The reviewer's note is well appreciated. The dynamic mechanical analyzer (DMA) has been utilized as a standard tool to characterize the hydrogel mechanical properties in many previous studies. DMA exerts oscillatory compressive or tensile force to evaluate soft or stiff materials. In determining the storage and loss moduli of the material, historical studies have displayed comparable DMA and rheometer results. For a linear isotropic Hookean material, Young's modulus of elasticity is considered equivalent to the compressive (or tensile) modulus.

Although the polymeric hydrogel is not a perfectly linear isotropic Hookean material, the linear, elastic region was first identified using a strain sweep in compression test and the storage modulus under oscillatory compression was determined in this linear region. We used this compressive modulus to represent the hydrogel stiffness in this study. As suggested, we have included literature hydrogel mechanical properties published previously for comparison.

Changes: “This provides control over the hydrogel mechanical properties, such as compressive modulus. Browning et al. reported this compositional control of PEGDA hydrogel stiffness with a compressive modulus range from 50 to 2500 kPa (13). Previous researchers have found that fewer bacteria adhere to soft PEGDA hydrogels than to stiff PEGDA hydrogels (14, 15).”

• P6, Line 20- Could the authors please clarify if the PAO1 pseudomonas is that a clinical strain or more of a “lab strain”

Response: Thank you and done. We have clarified that PAO1 is a lab strain.

Changes: “We used wild-type (WT) *P. aeruginosa* strain PAO1 and the mutant $\Delta pilY1$, also in the PAO1 background, in our experiments (16). PAO1 is a widely-used lab strain.”

• P9 Line 53-54- Why were two different PEG precursors used in this work? Could this be further clarified please. What was the effect on hydrogel mesh size, crosslinking ratio, surface/bulk chemistry. It is unclear why the same MW was not used to minimize potential differences between gels. The reviewer notes that the authors discuss mesh size later on in the manuscript, but it would be helpful to introduce this concept earlier.

Response: As mentioned in the introduction section, the molecular weight of the PEGDA macromer and the concentration can be used to modulate the hydrogel mechanical properties. In general, macromer molecular weight (e.g. 2kDa vs 10kDa) has a larger effect on stiffness than macromer concentration (e.g. 10% vs 20%). To obtain a substantial mechanical difference between the soft and stiff hydrogels, we used both macromer molecular weight and concentration. We have added the highlighted statement below to make this more clear. A smaller molecular weight PEGDA macromer typically results in higher crosslink density and smaller mesh size of the resultant hydrogel. Similarly, a higher macromer concentration typically increases the hydrogel crosslink density with a reduced hydrogel mesh size. As a result, the 50 wt% 2 kDa PEGDA hydrogels showed a much higher compressive modulus and smaller mesh size than the 10 wt% 10 kDa PEDA hydrogels. As noted by the reviewer, the authors recognized the potential effect of hydrogel mesh size on the nutrient and growth substrate transportation, and eventually, on the bacterial adhesion. We have previously decoupled these two variables with

the use of a 4-arm crosslinker and this approach could be used in future research studies to examine the effect of nutrient transport.

Changes: “Specifically, we made stiff hydrogels from 50 wt% 2 kDa PEGDA and we made soft hydrogels from 10 wt% 10 kDa PEGDA. These combinations of PEGDA molecular weight and crosslinker concentration were chosen to maximize the mechanical differences between the two gels.” We have also moved much of the discussion of mesh size to the Methods section.

- *It is unclear why the methods for calculating mesh size aren't provided in the methods section. There is no need to detail this in the results section.*

Response: Thank you and done.

Changes: We have moved the methods for calculating mesh size to the Methods section.

- *The authors should compare their calculated mesh size to that in literature.*

Response: Thank you and done.

Changes: “The estimated mesh size of our soft (~50 kPa) hydrogel was 8.3 nm and the estimated mesh size of our stiff (~4000 kPa) hydrogel was 0.9 nm, both well below the ~1 μm bacterial size. These are comparable to the mesh sizes previously measured for comparable PEGDA gels (13, 17, 18).”

Here, reference numbers are for the references at the end of this Response to Reviewers. Different references numbers are in the manuscript.

- *Figure 1 would be improved if both samples were provided on the same axis scale (log scale perhaps) to show the relationship between the two samples.*

Response: Thank you. The error bars cannot be seen well on the logarithmic y-axis, so we have added a second panel (Panel B) to this figure. Panel A allows the error bars to be seen, and Panel B shows the moduli of the two samples on the same logarithmic y-axis to show the relationship between them directly.

Changes: We have added a second panel to this figure (which is now Figure 2, due to the addition of a new Figure 1 following referee request). See below.

Fig. 2. Compressive moduli of the two PEGDA hydrogel formulations used, measured using a dynamic mechanical analyzer. (A) Because the values differ by two orders of magnitude, two different y-axes are used. Error bars show the standard error. (B) To show more clearly the relationship between the moduli of the two samples, both are plotted on the same logarithmic scale. For the softer gel, the error bars are too small to be seen on this scale. The modulus of the 10 wt% 10 kDa PEGDA gel was 44 ± 0.375 kPa and the modulus of the 50 wt% 2 kDa PEGDA gel was 3600 ± 560 kPa. N=6.

• Figure 2- Part b could be improved via a better caption to clarify the difference between WT reporter and WT Control and why they are on opposite sides of the graph.

Response: Panel B of this figure (which is now Figure 3) looked like that because we had formatted it clumsily – these are actually two different plots, using the same y-axis. We were trying to save space but it made the figure too difficult to read. We have split the graphs so that it is more clear that they are two graphs (because now there is a gap between them) and split this panel into two different panels, B and C. We have also re-labeled the other panels and adjusted the figure caption accordingly. We made additional minor formatting changes to this figure to try to make it read more easily. We made similar formatting changes to Figure 5.

Changes: We have reformatted this figure and Figure 5.

• Figure 3- An explanation of kurtosis could prove useful to readers that aren't as versed in the jargon of statistics.

Response: Thank you and done. We have added a further brief explanation of skewness and kurtosis in the main manuscript text, before the callout to Figure 4 B and C.

Changes: “The skewness values we measured were all positive, in agreement with our observation that measured distributions had a “tail” on the right (brighter) side of the mean; differences in the size of the skewness measure differences in how much of the distributions is found on the right (brighter) side of the mean. Kurtosis measures how much a distribution lies in the tail(s) and high kurtosis values corresponds to “heavy” tails, or having more of the distribution farther from the mean.”

- P14 Line 29-30- *“To examine how heterogeneous response might interplay with substrate stiffness, we measured the skewness and kurtosis of all attenuation- and metabolism-corrected brightness distributions - skewness to determine asymmetry and kurtosis to determine the preponderance of outliers (58). We found that, for all timepoints, the skewness and kurtosis were higher for populations attached to the softer hydrogel than for those attached to the stiffer hydrogel (Fig. 3 B and C). This showed that attachment to the softer gel was associated with more outliers and therefore, a more distinct subpopulation of “strong responders”. This statement is hard to understand/justify. The reviewer feels like a definitive conclusion to this issue can only be reached by looking at the genetics and not statistics alone. Could the authors please explain this further?*

Response: Thank you and done. We have added a clarification that “strong responder” here refers to the brightness of the GFP reporter which we use as our proxy measure for c-di-GMP concentration – i.e., this is meant to be a phenotypic, not a genotypic, description.

Changes: “This showed that attachment to the softer gel was associated with more outliers and therefore, a more distinct subpopulation of “strong responders” as measured by the brightness of the GFP reporter which we use as our proxy measure for c-di-GMP concentration. This does not indicate that the population on the soft gel has a stronger overall response to surface attachment, as shown by the average values in Figure 4A; rather, it indicates that a smaller fraction of the population responds strongly to attaching to a soft gel than responds strongly to attaching to a stiff gel. In contrast, attachment to the stiff gel was associated with fewer “strong responder” outliers and therefore a higher proportion of the population responding with increased c-di-GMP concentration.”

Reviewer: 2

Comments to the Author(s)

*In this manuscript, Blacutt et al. quantitatively characterized mechanobiological responses of *P. aeruginosa* on the stiffness of substrates for biofilm formation, using PEGDA hydrogels of higher and lower stiffness and confocal imaging. For rigorous fluorescent imaging-based evaluation, they calibrated attenuation of emitted fluorescent light through hydrogels and baseline metabolism of the tested cells. The authors' results clearly show differences in the fluorescence signal between the hard and soft hydrogel cases. They repeated their experiment using the bacteria strain not producing the envelope protein, and discussed a possibility for the bacteria cell could sense differences in pore size in addition to stiffness. The manuscript is well organized, and results are well presented. The conclusion is well supported by the presented data. I suggest minor revision with the following questions and suggestions.*

1. Line 39-41 in Page 1: It would be more rigorous to indicate that *E* values are approximate values by using \sim .

Response: Thank you and done. We have added this.

Changes: "It is well-established that *P. aeruginosa* cells increase intracellular levels of c-di-GMP after attaching to rigid solids such as glass, which has an elastic modulus of about \sim 20 GPa."

2. What is the gel thickness for bacteria culture? The thickness of hydrogel samples for swelling and stiffness measurement is shown clearly, but the gel thickness for bacteria culture is not easily found. It would be informative to clearly show the gel thickness of this case, because cells are known to sense the stiffness of the substrate or container below the gel (i.e., effective stiffness of hydrogel).

Response: Thank you for this. We have added additional information about how the gels were formed for imaging and how thick they were.

Changes: "Hydrogel Fabrication and Characterization

Hydrogels were prepared by first dissolving PEGDA in deionized water at a concentration of 10 wt% 10 kDa PEGDA or 50 wt% 2 kDa PEGDA. A photoinitiator solution (Irgacure 2959, 10 wt% in 70% ethanol) was then added at 1 vol% of the precursor solution. Imaging specimens were prepared by pipetting 4 μ L of the PEGDA solution into curing molds. The mold consists of an imaging spacer liner (Grace Bio-Labs SecureSeal™ Imaging Spacers) placed on a coverslip and sealed against a glass plate. Hydrogels were crosslinked by a 12-minute exposure to long wave UV light (Ultraviolet Products High Performance UV Transilluminator, 365 nm,

4mW/cm², Analytik Jena). The imaging spacers used each had a single well of diameter 13 mm and the liner, which was used as the mold for casting PEGDA gels, has a thickness of about 0.05 mm (Grace Bio-Labs, personal communication). Thus, the pre-swelling thickness of PEGDA gels used for imaging was about 0.03 mm. The adhesive spacers themselves were attached to the coverslip to enclose the gel after it was cast; these spacers have a thickness of 0.12 mm. Gels were then swollen to their equilibrium height by adding liquid medium. At the start of each imaging session the microscope objective was first focused on the coverslip bottom and then focused on the bacteria on the top of the gel. The height difference between these positions, read off the control software, gave an approximate measurement of gel thickness. Gels ranged from 0.1 mm to 0.13 mm in thickness.”

3. For the compressive modulus measurement part, it is recommended to include examples of the measured stress-strain curves in the SI or in Figure 1 (as insets) to indicate the strain range used for the modulus measurement.

Response: The authors appreciate this comment. We agree that examples of stress-strain curves would help the readers understanding the meaning of the mechanical data. Stress-strain curves of both hydrogels have been included as a new Figure 1, and all subsequent figures renumbered accordingly.

Changes: We show stress-strain curves as a new Figure 1 (see below).

Fig 1: Representative stress-strain curves of (A) 10 wt% 10 kDa PEGDA and (B) 50 wt% 2 kDa PEGDA hydrogel in compression. Because the values differ by two orders of magnitude, two different y-axes are used. The insets display the linear viscoelastic range of the curves used for

constant-strain frequency sweeps for each formulation. The average storage modulus at 1.25 Hz was 44 ± 0.375 kPa for the 10 wt% 10 kDa PEGDA gel and 3600 ± 560 kPa for the 50 wt% 2 kDa PEGDA gel. N=6.

4. One hour was allowed for bacteria to adhere to hydrogel, and I wonder how this time duration was determined. Could imaging be done 0.5 hour after the inoculation of hydrogels?

Response: It took a little less than an hour to identify the 10-15 sites containing adhered bacteria that would be subsequently monitored by timelapse microscopy. To keep the timestep between introducing bacteria to the substrate and beginning the timelapse microscopy the same across all experiments, we chose 1 hour as the time to allow adhesion.

Changes: We have clarified this by adjusting the highlighted text:

“For all experiments, we used an Olympus FV1000 motorized inverted IX81 microscope suite, with instrument computer running FV10-ASW version 4.2b software, to image attached bacteria using laser-scanning confocal microscopy. To prepare the bacteria, we first diluted 40 μ L of an overnight culture into 5 mL of fresh LB media containing gentamicin. We then placed an imaging spacer (Grace Bio-Labs SecureSeal™ Imaging Spacers) on both the microscope slide and coverslip around the PEGDA hydrogel. 25 μ L of the bacterial dilution was inoculated onto the PEGDA hydrogel substrate on a glass coverslip and sealed to a microscope slide. The slide was then placed on the microscope stage and bacteria were allowed to adhere to the hydrogel for an hour prior to imaging. During this hour 10-15 locations containing adhered bacteria were identified for subsequent time-series imaging. Imaging was done using a 60x oil-immersion objective, a 488-nm laser with a 405/488 excitation filter, and an emission filter of 505/605. For each day’s worth of experiments, 10-15 sites were imaged every 30 minutes for a total of 3 hours. This process was repeated on three different days for each condition. To image each site a confocal Z-stack was taken with a depth of 6 μ m and an inter-slice size of 750 nm.”

5. The last line in Page 8: How many cells were typically imaged at each time point? Since probability distribution curves are shown in the manuscript, providing this number (rough estimate or range) seems required for supporting the reliability of the curves.

Response: Thank you for this suggestion. We have done new analysis for this and added the results in the following highlighted statement.

Changes: “For each day’s worth of experiments, 10-15 sites were imaged every 30 minutes for a total of 3 hours. This process was repeated on three different days for each condition. To image each site a confocal Z-stack was taken with a depth of 6 μm and an inter-slice size of 750 nm. At each site on each day, roughly 40-90 bacterial cells were imaged at each initial timepoint and roughly 80-160 bacterial cells were imaged at each final timepoint. “

We have also added Zilei (Eva) Chen and Mara Eccles, undergraduates who are working on a project analyzing the accumulation of bacteria on these gels over time, to the acknowledgements of this paper because their analysis was used to determine these bacterial counts.

6. *Figures 2A & 4A: It would be informative to provide the time point and gel type (hard or soft) for the shown image.*

Response: Thank you and done.

Changes:

“**Fig. 3.** Imaging and calibration for WT PAO1. (A) Representative micrograph of GFP-expressing *P. aeruginosa*, attached to the soft gel substrate at the first imaging timepoint.”

“**Fig. 5.** (A) Example image of attached $\Delta pily1$ cells, on the soft gel at the first imaging timepoint.”

7. *Figure 2C: I just wonder why the curves are not symmetric and why they do not collapse even after correction. Are these due to any possible limitations of the authors’ method? Then, don’t they have to be considered in analyzing images of bacteria?*

Response: This is a good question. We don’t know why the curves are not symmetric nor why the collapse onto each other once scaled does not result in a perfect match. The asymmetry likely corresponds to an asymmetric distribution of bead brightnesses, and perhaps also is impacted by lensing or attenuation artefacts caused by internal heterogeneities in the gel substrates. The differences in the distribution shapes, shown by

the collapse not resulting in a perfect match, likely arise from such heterogeneity-caused optical artefacts.

We think the reviewer makes a good point that this shows a limitation of our method, which needs to be held in mind. We have added a brief section just before the Summary and Conclusion summarizing these issues.

Changes: **“Possible effects of gel heterogeneity**

Figure 3D shows brightness distributions for fluorescent beads that are both asymmetric and do not perfectly collapse onto each other when scaled. The causes for these phenomena, which may suggest a limit to the method we present here, are not known. The asymmetric distributions likely corresponds to an asymmetric distribution of bead brightnesses, and perhaps also is impacted by lensing or attenuation artefacts caused by internal heterogeneities in the gel substrates. The differences in the distribution shapes, shown by the collapse not resulting in a perfect match, likely arise from such heterogeneity-caused optical artefacts. The asymmetries and shape differences seen for beads are much smaller than the ones we measure for bacterial populations on the same gels, so the overall conclusions of this work and the method shown here are not affected. However, future work using higher-resolution, more fine-grained signaling measurements may need to account for internal heterogeneities in gel substrates.”

8. *How was the attenuation factor of 0.662 shown in Page 12 determined? Is it just a ratio of the mean values?*

Response: Yes. We have added a statement clarifying this.

Changes: **“We then calculated an attenuation factor of 0.662 from the ratio of the mean value of the two populations (beads on soft hydrogels to beads on stiff hydrogels).”**

9. *Figure 3A: The brightness level of the soft gel case became higher than that of the hard gel between ~140 min and 200 min. What does this mean?*

Response: This is also a good question. We don't know why the brightness level of bacteria on the soft gel became higher in this time window than the brightness level of bacteria on the stiff gel. This is what led us to speculate that factors other than modulus per se (e.g., mesh size) may also be playing a role in our GFP reporter readout.

We have drawn the connection more clearly by adding the following statement:

Changes: “Thus, bacteria on the softer hydrogel could have access to a larger effective volume of growth substrate than will bacteria on the stiffer hydrogel, which could impact both metabolism and c-di-GMP signaling. Furthermore, the impact of mesh size on access to growth substrate and therefore on metabolism might be a reason that WT bacteria become brighter on the soft gel than on the hard gel between ~140 and ~200 minutes (Fig. 4A). We will investigate the effect of mesh size on bacterial response to attachment further in later work.”

10. Line 38-45 in Page 14: What are “strong responders” exactly? Are they related to increased c-di-GMP concentration (“strong” suggests this correlation)? If so, is more strong responders on the soft gel contradict to the higher population of bacteria with increased c-di-GMP concentration? I felt this part is confusing.

Response: Thank you for pointing out this point of possible confusion. We have added language (highlighted) to clear this up.

Changes: “This showed that attachment to the softer gel was associated with more outliers and therefore, a more distinct subpopulation of “strong responders” as measured by the brightness of the GFP reporter which we use as our proxy measure for c-di-GMP concentration. This does not indicate that the population on the soft gel has a stronger overall response to surface attachment, as shown by the average values in Figure 4A; rather, it indicates that a smaller fraction of the population responds strongly to attaching to a soft gel than responds strongly to attaching to a stiff gel. In contrast, attachment to the stiff gel was associated with fewer “strong responder” outliers and therefore a higher proportion of the population responding with increased c-di-GMP concentration. These results suggested that substrate mechanics might impact the development of heterogeneity in populations of surface-attached bacteria.”

11. Repeatability: Figures 3 and 5 show error bars, and I wonder if these error bars are standard deviation from repeated experiments. The bottom of Page 8 shows that the whole process of imaging was repeated on three different days. Does this mean that the n values for the graphs in Figures 3 and 5 are 3? I believe it is very important to clearly explain how repeatable the authors’ experiment was.

Response: Thank you for pointing this out. We have added the following statement to both of the figure captions:

Changes: “Each data point shows the mean value of 3 independent biological replicates (N=3), with error bars the standard error of the mean.”

In this sense, I wonder whether the attenuation factor shown in Page 12 was measured repeatedly and whether the authors got similar values as 0.662. This is also important because similar attenuation factor values obtained from repeated experiments would show that the authors' method is repeatable.

Response: Thank you for pointing this out. We have added the following highlighted text to the Methods section.

Changes: All beads were imaged using the same laser, but different intensity, photomultiplier, and image acquisition settings; these were different settings to ensure the beads were not over exposed in our images. These measurements were done on three different days, with different gel preparations on each day. The numbers of beads measured on each day were: 995 beads, 1769 beads, and 1407 beads on the soft gels, and 262 beads, 1752 beads, and 1285 beads on the stiff gels. Similar attenuation factors were measured for each day.

Also, I wonder how repeatable the average fluorescence intensity of control bacteria was. If this were not repeatable (similar between repeated experiments), the authors' way of removing the baseline metabolism level could be problematic.

Response: Thank you for pointing this out. The biggest variability was found among the dimmest of the control bacteria, because these were near the lowest brightness measurable using our confocal microscope settings. The peaks of the brightness distributions (near the mean value, which is what we used for metabolism correction) were far less variable. Also, the trends over time (i.e. the pattern of fluorescence timecourses) were consistent between replicates. Thus, the time-dependence of metabolism changes (measured using GFP brightness) was consistent and therefore we feel confident in this control method.

Changes: “This removed fluorescence resulting from the metabolism of these cells and leaves only the fluorescence associated with intracellular c-di-GMP levels (Fig 3 E). This approach is rooted in that used for the original development and validation of the reporter plasmid, and its subsequent use (19-22). We used a similar approach in our prior work on bacterial mechanosensing (23).”

The reference numbers above refer to the references found at the end of this Response to Reviewers; reference numbers in the manuscript are different.

===PREPARING YOUR MANUSCRIPT===

- *one version identifying all the changes that have been made (for instance, in coloured highlight, in bold text, or tracked changes);*
- *a 'clean' version of the new manuscript that incorporates the changes made, but does not highlight them. This version will be used for typesetting.*

References cited in this Response to Referees

1. Rodesney CA, Roman B, Dhamani N, Cooley BJ, Katira P, Touhami A, et al. Mechanosensing of shear by *Pseudomonas aeruginosa* leads to increased levels of the cyclic-di-GMP signal initiating biofilm development. *Proceedings of the National Academy of Sciences of the USA*. 2017;114(23):5906-11.
2. Siryaporn A, Kuchma S, O'Toole G, Gitai Z. Surface attachment induces *Pseudomonas aeruginosa* virulence. *Proceedings of the National Academy of Sciences of the USA*. 2014;111(47):16860-5.
3. Jenal U, Reinders A, Lori C. Cyclic di-GMP: second messenger extraordinaire. *Nature Reviews Microbiology*. 2017;15(5):271-84.
4. Baker AE, Webster SS, Diepold A, Kuchma SL, Bordeleau E, Armitage JP, et al. Flagellar Stators Stimulate c-di-GMP Production by *Pseudomonas aeruginosa*. *Journal of Bacteriology*. 2019;201(18):e00741-18.
5. Galperin MY. What bacteria want. *Environmental Microbiology*. 2018;20(12):4221-9.

6. Hall CW, Mah T-F. Molecular mechanisms of biofilm-based antibiotic resistance and tolerance in pathogenic bacteria. *FEMS Microbiology Reviews*. 2017;41(3):276-301.
7. Meylan S, Andrews IW, Collins JJ. Targeting Antibiotic Tolerance, Pathogen by Pathogen. *Cell*. 2018;172(6):1228-38.
8. Ciofu O, Tolker-Nielsen T. Tolerance and Resistance of *Pseudomonas aeruginosa* Biofilms to Antimicrobial Agents—How *P. aeruginosa* Can Escape Antibiotics. *Frontiers in Microbiology*. 2019;10(913).
9. Nachlas ALY, Li S, Jha R, Singh M, Xu C, Davis ME. Human iPSC-derived mesenchymal stem cells encapsulated in PEGDA hydrogels mature into valve interstitial-like cells. *Acta Biomaterialia*. 2018;71:235-46.
10. Zhai X, Ruan C, Ma Y, Cheng D, Wu M, Liu W, et al. 3D-Bioprinted Osteoblast-Laden Nanocomposite Hydrogel Constructs with Induced Microenvironments Promote Cell Viability, Differentiation, and Osteogenesis both In Vitro and In Vivo. *Advanced Science*. 2018;5(3):1700550.
11. Lim WS, Chen K, Chong TW, Xiong GM, Birch WR, Pan J, et al. A bilayer swellable drug-eluting ureteric stent: Localized drug delivery to treat urothelial diseases. *Biomaterials*. 2018;165:25-38.
12. Choi JR, Yong KW, Choi JY, Cowie AC. Recent advances in photo-crosslinkable hydrogels for biomedical applications. *BioTechniques*. 2019;66(1):40-53.
13. Browning MB, Wilems T, Hahn M, Cosgriff-Hernandez E. Compositional control of poly(ethylene glycol) hydrogel modulus independent of mesh size. *Journal of Biomedical Materials Research Part A*. 2011;98A(2):268-73.
14. Kolewe K, Zhu J, Mako N, Nonnenmann S, Schiffman J. Bacterial Adhesion Is Affected by the Thickness and Stiffness of Poly(ethylene glycol) Hydrogels. *ACS Applied Materials and Interfaces*. 2018;10:2275-81.
15. Kolewe KW, Peyton SR, Schiffman JD. Fewer Bacteria Adhere to Softer Hydrogels. *ACS Appl Mater Interfaces*. 2015;7(35):19562-9.
16. Jacobs MA, Alwood A, Thaipisuttikul I, Spencer D, Haugen E, Ernst S, et al. Comprehensive transposon mutant library of *Pseudomonas aeruginosa*. *Proceedings of the National Academy of Sciences*. 2003;100(24):14339-44.
17. Hagel V, Haraszti T, Boehm H. Diffusion and interaction in PEG-DA hydrogels. *Biointerphases*. 2013;8(1):36.
18. Liao H, Munoz-Pinto D, Qu X, Hou Y, Grunlan MA, Hahn MS. Influence of hydrogel mechanical properties and mesh size on vocal fold fibroblast extracellular matrix production and phenotype. *Acta Biomaterialia*. 2008;4(5):1161-71.
19. Rybtke MT, Borlee BR, Murakami K, Irie Y, Hentzer M, Nielsen TE, et al. Fluorescence-based reporter for gauging cyclic Di-GMP levels in *Pseudomonas aeruginosa*. *Appl Environ Microbiol*. 2012;78:5060-9.
20. Irie Y, Borlee BR, O'Connor JR, Hill PJ, Harwood CS, Wozniak DJ, et al. Self-produced exopolysaccharide is a signal that stimulates biofilm formation in *Pseudomonas aeruginosa*. *Proceedings of the National Academy of Sciences of the United States of America*. 2012;109:20632-6.
21. Moscoso JA, Jaeger T, Valentini M, Hui K, Jenal U, Filloux A. The Diguanylate Cyclase SadC Is a Central Player in Gac/Rsm-Mediated Biofilm Formation in *Pseudomonas aeruginosa*. *Journal of Bacteriology*. 2014;196(23):4081-8.
22. Chua SL, Tan SY-Y, Rybtke MT, Chen Y, Rice SA, Kjelleberg S, et al. Bis-(3'-5')-Cyclic Dimeric GMP Regulates Antimicrobial Peptide Resistance in *Pseudomonas aeruginosa*. *Antimicrobial Agents and Chemotherapy*. 2013;57(5):2066-75.

23. Rodesney CA, Roman B, Dhamani N, Cooley BJ, Touhami A, Gordon VD. Mechanosensing of shear by *Pseudomonas aeruginosa* leads to increased levels of the cyclic-di-GMP signal initiating biofilm development. *Proceedings of the National Academy of Sciences*. 2017;114:5906-11.